# A comparative study of the cortical function during the interpretation of algorithms in pseudocode and the solution of first-order algebraic equations

Oscar Hernández[1]*, Eduardo Zurek[2], John Barbosa[2], Minaya Villasana[3]

1 Departamento de Química y Biología, Universidad del Norte, Barranquilla, Atlántico, Colombia,
2 Departamento de Ingeniería de sistemas, Universidad del Norte, Barranquilla, Atlántico, Colombia,
3 Departamento de Cómputo Científico y Estadística, Universidad Simón Bolivar, Caracas, Venezuela

* ohernandezb@uninorte.edu

**Data Availability Statement:** The data are available at: https://github.com/oscarehb/EEG-data-during-

## Abstract

This study intends to determine whether similarities of the functioning of the cerebral cortex exist, modeled as a graph, during the execution of mathematical tasks and programming related tasks. The comparison is done using network parameters and during the development of computer programming tasks and the solution of first-order algebraic equations. For that purpose, electroencephalographic recordings (EEG) were made with a volunteer group of 16 students of systems engineering of Universidad del Norte in Colombia, while they were performing computer programming tasks and solving first-order algebraic equations with three levels of difficulty. Then, based on the Synchronization Likelihood method, graph models of functional cortical networks were developed, whose parameters of Small-World-ness (SWN), global($E_g$) and local ($E_l$) efficiency were compared between both types of tasks. From this study, it can be highlighted, first, the novelty of studying cortical function during the solution of algebraic equations and during programming tasks; second, significant differences between both types of tasks observed only in the delta and theta bands. Likewise, the differences between simpler mathematical tasks with the other levels in both types of tasks; third, the Brodmann areas 21 and 42, associated with auditory sensory processing, can be considered as differentiating elements of programming tasks; as well as Brodmann area 8, during equation solving.

## Introduction

In a study carried out by White G and Sivitanides M. in 2002 [1], it was observed that university students who had a good performance in first-year mathematic courses, had the necessary cognitive characteristics to perform equally well on second-year visual programming classes. Their research further shows that analytical and logical thinking skills are necessary to perform with success, not only learning mathematics, but in learning procedural programming language, object-oriented programming languages and visual programming languages. Following this approach, in [2] it is observed, with an example, that algorithmic and mathematical

the-interpretation-of-algorithms-in-pseudocode-and-the-solution-of-equation.

**Funding:** The author(s) received no specific funding for this work.

**Competing interests:** The authors have declared that no competing interests exist.

thinking complement each other, and that an algorithmic approach leads to questions that encourage students to go deeper into mathematics. Therefore, there is an interrelation between algorithmic and logical thinking required for mathematics. The hypothesis raised in this study is that both type of tasks must give rise to cortical networks or graphs with similar characteristics.

Other studies show, from the teaching-learning processes, a relationship between computational and mathematical thinking [1–3]. In Computational or mathematical thinking, other processes intervene as well, such as reading and writing [4]. In this study, two types of tasks whose respective solution involves a high component of logical thinking processes, were compared, since, on the one hand, the solution of equations requires an understanding of the relationship between the variables, and on the other hand, the analysis of the algorithm functions requires the interpretation of order relations between the different steps that are necessary to solve a problem.

The anatomic and physiological characteristics of the cerebral cortex suggest that it can be divided in nodes, or functional or anatomical units, which are connected to each other depending on the sensory modality that they process and, on the state (e.g., normal, pathological, sleep, wakefulness) they are in [5–10]. In general, it can be inferred that it works as a Small World Network, that is: a complex network with multiple space-time scales, that supports segregated (high grouping) and distributed (short path length) information processing, which works maximizing the efficiency and minimizing information processing costs (high global efficiency at a low cost), and supports instant parallel processing.

The computational exploration of cerebral cortex functioning can be made in different ways among which are: by simulation of each neuron and their interrelations, or from the development of graphs that model the structure or function of a cortical network. The latter is based on the analysis of recorded signals, for example, from electroencephalography (EEG) or functional magnetic imaging (fMRI) [11, 12]. In this study, graphs that model the cortical networks were constructed, in which nodes represented the electrodes of the EEG equipment and edges in the graph represented the interaction between the different brain centers [11, 13, 14].

Network or graph models made from EEG signals depend on the number of electrodes or channels, thus this method reduces model complexity and the need for extensive computational resources. The cortical network models developed from fMRI images tend to be more complex since the number of nodes increases and their identification is visual [11]. In general, from EEG based models, functional differences can be observed between brains in different states.

To the best of the authors' knowledge, there are no known previous studies that link both types of thinking from physiological indicators, nor are there studies that link patterns of cortical connectivity during the two scenarios: while performing mathematical tasks and computer programming tasks. Therefore here, for the first time network cortical models (graphs) based on EEG are compared during the execution of mathematical and programming tasks. Also, it is important to highlight the exploration of cortical functioning using three levels of complexity and not two, as has been done in previous studies on basic arithmetic operations (multiplications) [15].

This work stems from the statement that algorithmic thinking is intimately related to mathematical thinking, thus it can be speculated that these tasks may give rise to cortical networks that share similarities in terms of the measured parameters of the cortical networks such as global and local efficiencies and smallworldness. To assess this speculation, we propose the measurement of these parameters while mathematical and programming tasks were performed to determine its characteristics, similarities, and differences between them while performing mathematical tasks and algorithmic tasks with varying levels of difficulty.

## Materials and methods

### Participants

Electroencephalographic recordings were made with 16 volunteer students from 5th semester of the systems engineering program from Universidad del Norte (Barranquilla- Colombia). All the participants met the following selection criteria: students that dominate the programming requisites for the tasks which is typically attained after the 4th semester of System's engenieering thus inducing an age range between 18 and 21 years, with normal or corrected vision, right-handed, not having consumed (prior to recordings) coffee, tea or other nervous system stimulant, not having drunk alcohol in the 48 hours prior to test day. Age distribution for the 16 subject sample used in this study is: 1 participant was 18 years old, 9 were 19 years old, while 3 participants were 20 and 21 years old respectively. The mean age is 19.5 and the gender distribution was 2 female and 14 male subjects.

### Encephalographic recordings and protocol for tasks performance

The EEG records were obtained for each participant during the execution of six tasks: mathematical tasks with three difficulty levels, low (ES task), medium (EM task) and high (EC task), and computer programming tasks for low (PS task), medium (PM task) and high (PC task) difficulty levels. Each task consisted of a set of 20 stimuli (problems). Two further tasks were added as control experiments: a subitizing task (Do) in which 2 to 4 random points were projected for 20 seconds, and a task (Cr) in which a white cross was projected on a black background for 30 seconds. All of these tasks were under the same experimental protocol (see Fig 1), applied from PsychoPy [16–18], which is an open multi-platform software for experiments in neuroscience and experimental psychology. The data was recorded using WinEEG software [19] and the tasks were projected for each subject using PsychoPy. Ocular correction and artefact rejection was performed through the ICA from the EEGLAB software. The signal adjustments from the EEG tick marks were performed in Matlab with in-house programming.

The mathematical tasks were classified by difficulty according to the number of operations required to solve each equation. The computer programming tasks consisted of analyzing an algorithm and determining whether it fulfilled the function expressed in the input—output relationship. The complexity was defined according to the number of code lines plus the input-output statements. In this way, low-level tasks (PS *task*) had 4 to 6 lines, medium-level tasks (PM *task*) had between 7 and 8, and high-level tasks (PC *task*) had 9 to 12. The volunteers in this study are skilled in programming, thus reading algorithms in pseudocode is an acquired competency as is solving first degree algebraic equations. Fig 1 shows the workflow of the experiments conducted with each participant as well as examples of each level of difficulty.

The electroencephalography equipment used was a *Mitsar EEG* 201 [19] with 19 electrodes located on the scalp according to the international 10–20 system (see Fig 1). These were: Fp1, Fp2, F7, F3, Fz, F4, F8, T3, C3, Cz, C4, T4, T5, P3, Pz, P4, T6, O1, O2, referenced to the earlobes. Each electrode and their reference define channels from Ch01 to Ch19 respectively, as shown in Fig 2.

Recordings were made at a rate of 250 Hz in a soundproof camera. Participants were seated in a comfortable chair and the stimuli presented on a laptop computer screen located 70 cm away from their face. From these recordings, models (graphs) were constructed, and their corresponding parameters calculated (SWN, $E_g$ and $E_l$ for each node or channel). In all cases, the calculations were made per subject, task and complexity level for each of the frequency bands defined in [20], modifying only the lower limit of the delta band as per the range allowed by

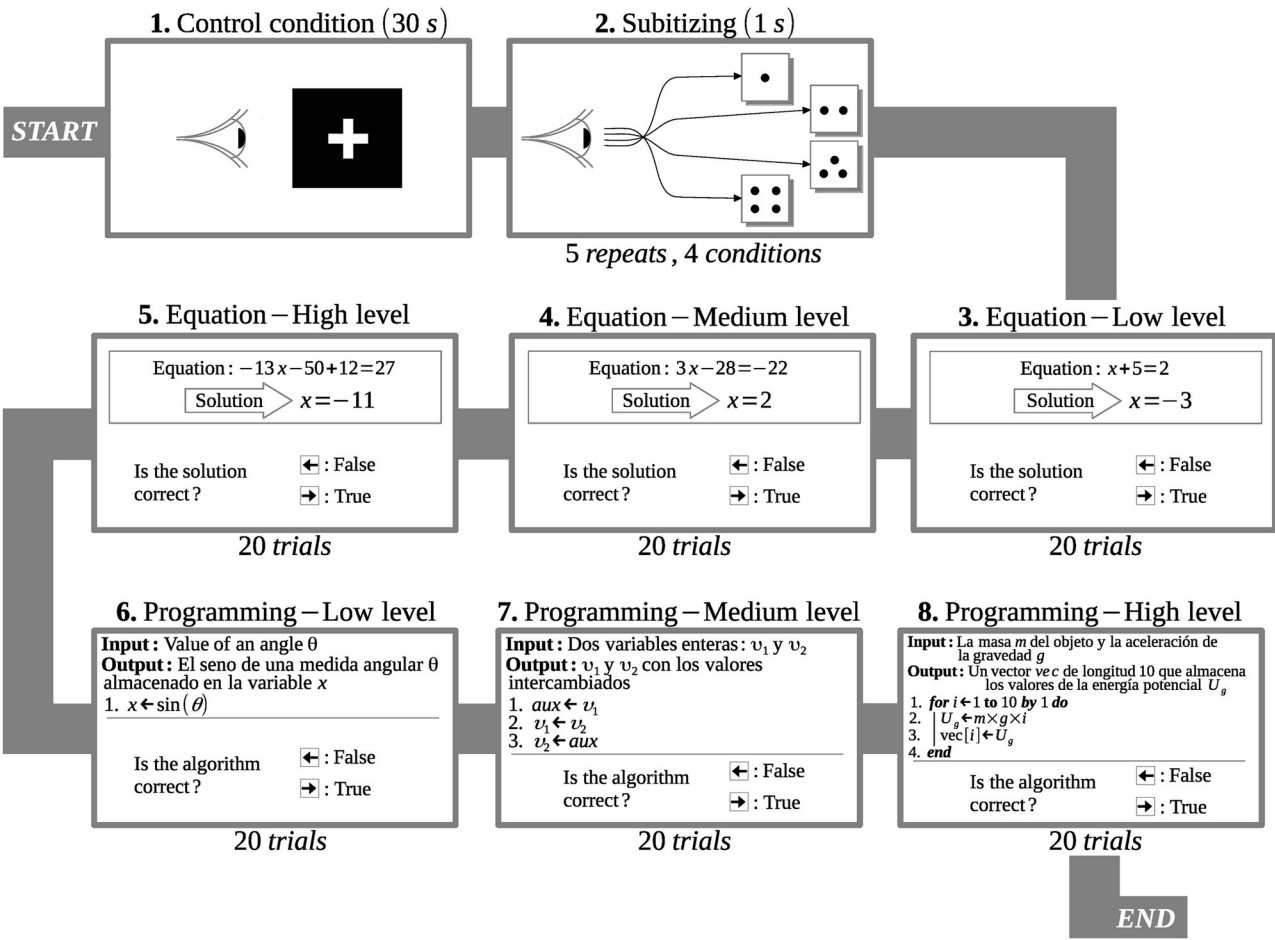

**Fig 1. Experimental protocol.** Experimental protocol for carrying out tasks during EEG data collection.

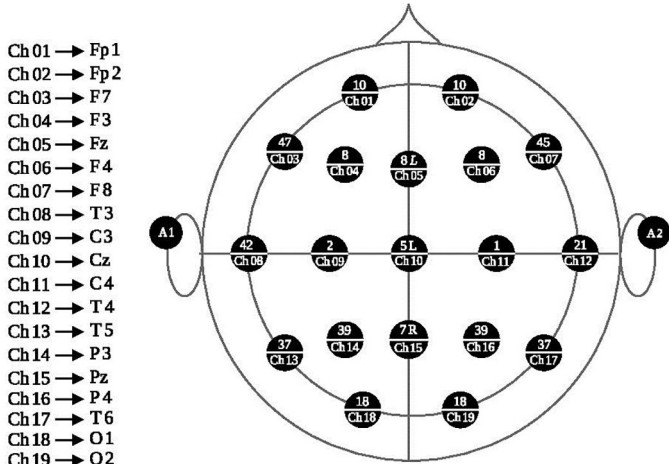

**Fig 2. Spatial localization of EEG electrodes.** Spatial localization of EEG electrodes, the top number on each electrode is the Broadmann area for that electrode.

the equipment used: delta (0.01 to 4 Hz), theta (4 to 8 Hz), alpha (8 to 12 Hz), beta (12 to 30 Hz) and gamma (30 to 50 Hz).

## Synchronization likelihood

For the development of cortical network models from EEG recordings, it is necessary to determine whether or not the signals are related. This can be achieved by estimating the synchronization between the signals coming from each channel. There are various methods to detect this synchronization. In this study, the SL index (*Synchronization Likelihood*) was used as a measurement of the generalized synchronization [21, 22]. The SL index describes how strong is the synchronization of channel k with all other channels, at time i. In this study, the HERMES package was used [23] in order to generate the matrices with SL indices of the EEG signals. HERMES can be downloaded for free from its website [24].

## Graph analysis

The interconnectedness of the various channels can be represented by a graph, which is a data structure representing a group of objects called nodes (here the nodes of the graphs are the channels), joined together by links or edges that represent their interrelations. Graphs can be characterized by various metrics or parameters. Among the graph parameters commonly used in neuroscience to describe the structure and the interrelatedness of the nodal activity are: the node degree; node strength; the distributed degree of a node and the distributed power law; the clustering coefficient of a node; the average clustering coefficient of the graph; the geodesic (for a node); the characteristic path length, the Small-Worldness, and local and global efficiencies [25–34]. This paper is focused on three graph parameters as a way to characterize cortex activity and provide information on patterns exhibited on different Brodmann zones through the execution of the various tasks. Thus the 3 parameters considered here are: Small-Worldness (SWN), global efficiency ($E_g$), and local efficiencies ($E_l$) as defined in [34]. To define those parameters, two additional metrics of the graph are defined: characteristic path length and the clustering coefficient.

*Characteristic path length* (L): is defined as the average of geodesics among all network node pairs (see Eq (1)).

$$L = \frac{1}{n}\sum_i L_i = \frac{1}{n(n-1)}\sum_i \sum_{j \neq i} d_{ji},$$

(1)

where $L_i$ is the average distance between node i and the other nodes and $d_{ji}$ is the geodesic or the shortest path between nodes i and j, that is, the minimal number of links to get from node i to node j in the graph structure.

*Clustering coefficient*: (C): measures the degree of connectivity of the network or the tendency of the nodes to group together (see Eq (2)).

$$C = \frac{1}{n}\sum_i C_i = \frac{1}{n}\sum_i \frac{t_i}{k_i\frac{(k_i-1)}{2}},$$

(2)

where $C_i$ is the clustering coefficient for node i, $k_i$ is the degree of node i (or number of connections to other nodes), and $t_i$ is the number of triangles connected to this node (cardinality of a 3-clique).

*Small-Worldness* (SWN): measures the degree in which a network tends to be a Small World Network (see Eq (3)). Small World Networks can result from reconnecting some distant nodes of a regular network [28, 35], and are characterized by having a larger C and an L

similar to that of a random network.

$$\text{SWN} = \frac{C/C_{\text{rand}}}{L/L_{\text{rand}}} = \frac{C \cdot L_{\text{rand}}}{L \cdot C_{\text{rand}}}.$$ (3)

In Eq (3), $C_{\text{rand}}$ and $L_{\text{rand}}$ are the clustering coefficient and the characteristic path length of a random network. The random graphs were created using the HERMES software which states: "The graph created is entirely random, with the probability of connection tested for each pair of vertices (Erdos-Renyi random graph). Only single edges and no self-loops are allowed—hence, P is adjusted to account for the slight reduction in the number of maximum edges". In a random network L and C decrease with respect to the regular and Small World Networks [28]. A regular network is the result of connecting each node with their nearest neighbors. The regular networks have larger L and C than Small World Networks [28].

For Small World Networks, $C/C_{\text{rand}}$ is greater than 1 and $L/L_{\text{rand}}$ is approximately equal to 1. Therefore, SWN can be regarded as a measurement of the degree in which the graph (cortical activity) tends to be a Small World Network.

The following references may help readers with a better understanding of this metric and its importance for network characteristics: [35–37].

*Global efficiency* ($E_g$): the global efficiency of a network, is the efficiency with which the network exchanges information within the nodes as in Eq (4).

$$E_g = \frac{1}{n}\sum_i E_{li} = \frac{1}{n(n-1)}\sum_i \sum_{j\neq i}\frac{1}{d_{ji}}.$$ (4)

*Local efficiency* ($E_l$): is the efficiency with which a node interchanges information with the surrounding nodes. This metric is defined in Eq (5).

$$E_{li} = \frac{1}{n(n-1)}\sum_{j\neq i}\frac{1}{d_{ji}}.$$ (5)

## Statistical analysis

The Shapiro Wilk test was used to determine if the data fit a Normal distribution. The result showed that, in most cases, the data distribution for SWN and $E_g$ did not correspond to a Normal distribution. Therefore, the differences between the different pairs of task types were evaluated using the two-tailed Mann & Witney asymptotic U-test. The Supporting information (see S6 Table) also includes results from Friedman test to attest the statistical differences between the 3 difficulty levels complementary to the U-Mann Whitney test.

Boxplots were used for descriptive purposes of SWN and $E_g$ values for each type of task and level of complexity. In these, in addition to the median, mean values were also depicted. Likewise, average values of the local efficiencies for each channel and type of task, were represented with heat maps. Subsequently, to reduce the dimensionality of local efficiency data corresponding to each channel and frequency band for different levels of difficulty in both types of tasks, cluster analysis was carried out. In this way differences between the different levels of difficulty and task type were observed over the various channels.

## Results and discussion

Experiments were carried out according to the experimental setup described in section Materials and methods. Subjects performed the tasks and recordings were made. Signal

synchronization was assessed by computing the SL index using HERMES, and the resulting graph was constructed for each subject and task. The 3 main graph parameters defined above were computed and the results from each case follows.

## Small-worldness

Fig 3 panels B-F provide insight on the distribution of SWN values for the different types of tasks in each frequency band. Fig 3A depicts the median values in panels B-F. The vertical gray lines separate the diagrams for each type of task: mathematical tasks (simple, medium, and complex), programming tasks (simple, medium, and complex), the control condition (Cr), and the subitizing task (Do), thus making it easier to visually identify possible differences between the tasks. Panel A groups the median values for the tasks in each band, while the mean value in each case is shown with a diamond.

In Fig 3, the lowest values for SWN (very close to 1) are observed in the delta band (marked with an x on panel A), while the highest SWN values are seen in the beta band (marked with a triangle on panel A).

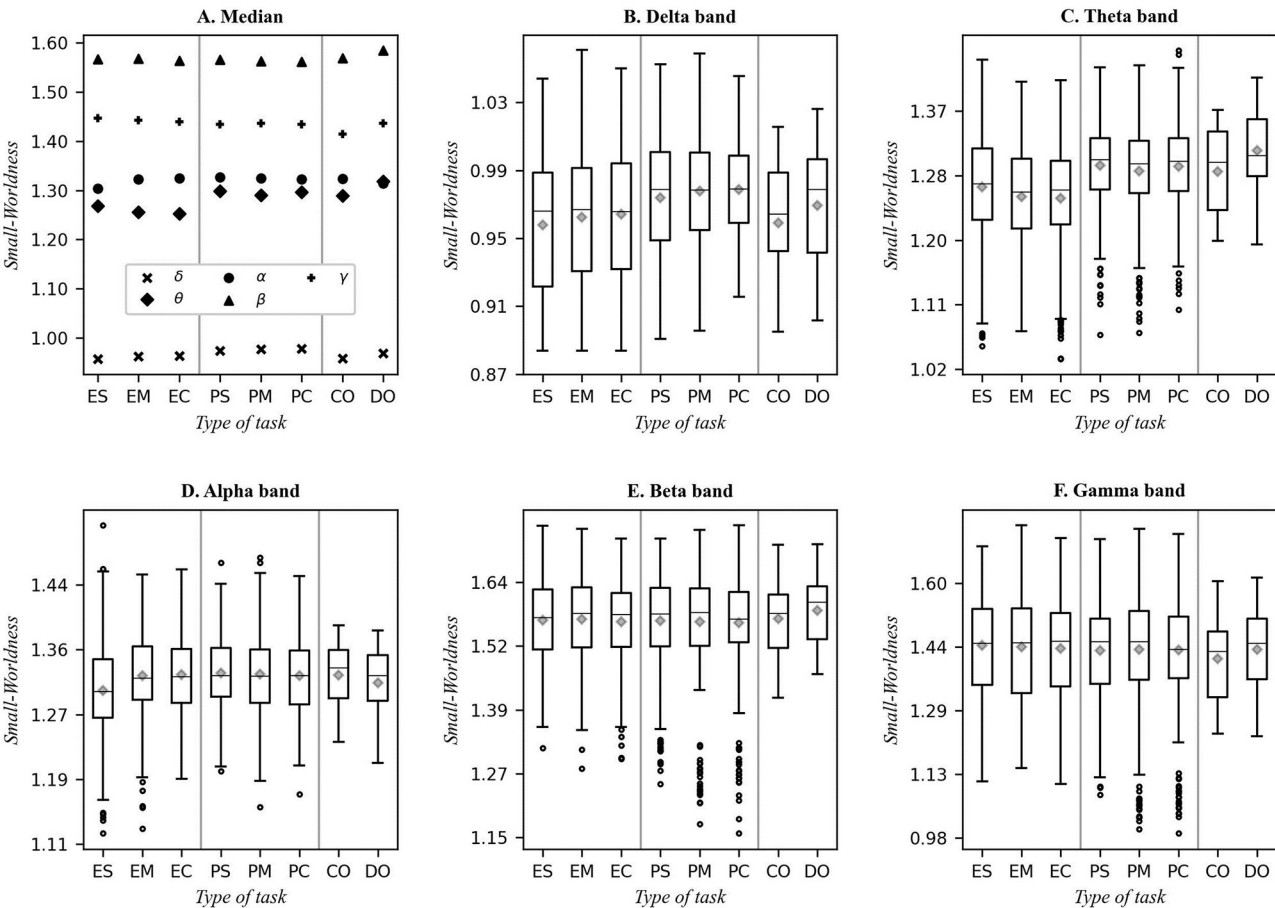

**Fig 3. Small-Worldness for the delta, theta, alpha, beta and gamma frequency bands.** In A, the medians are compared. In B-F, the boxplots for each band are shown. The gray lines separate the tasks in three parts: simple mathematical tasks (ES), medium complexity tasks (EM) and complex tasks (EC); also shown are the programming tasks in the three previous levels (PS, PM and PC), as well as control condition (Cr) and subitizing (Do). Besides the median (horizontal line), the mean is shown with a diamond on each case.

High values for SWN in the beta band is in accordance with the results of a recent study which shows that beta waves are the mechanism of control which allows for the conscious exchange between different pieces of information contained in the working memory [38]. Therefore, the highest SWN values for the beta band suggests a close relationship between this coefficient and working memory. The existence of a SWN-Working memory relationship has been previously established [39, 40]. For equation solving and programming tasks, regardless of the complexity level, the use of working memory is essential, this might explain the reasons for the small differences in the beta band (Fig 3E) between the different levels of complexity [38]. Using an asymptotic 2-tailed Mann-Whitney test, it is seen that p-values are large compared to the significance level 0.05 (please refer to the supporting information for the complete tables on p-values, see from S1 to S5 Tables), thus there is no evidence to reject the null hypothesis of equal means.

The bands that showed difference between the mathematical and the programming tasks are: delta, theta and alpha to a lesser extent (see Fig 3A–3D). These differences are statistically significant according to the 2-tailed UMann-Whitney tests, p-values are all below the significance level on the table for the delta and theta bands when comparing programming activities to mathematical tasks in supporting information (see from S1 to S3 Tables). However, no differences are observed within the same type of task in the delta band. On the theta band this holds true for simpler mathematical tasks compared to all other tasks. On the alpha band only the simpler mathematical tasks show a statistically different mean on SWN compared to all other tasks (p-values in S3 Table are all less than 0.05).

Fig 3A reveals a reduction in the cortical network connectivity in the delta band. This can be due to a reduction in the number of edges (L is reduced) that connect both remote brain regions and nearby regions. Note that the minimal theoretical value of SWN is 1, however in Fig 3B, despite being very close, SWN values drop below 1, which can be due to errors in the electroencephalography recording and/or rounding errors.

## Global efficiency

Fig 4B–4F show the distribution for global efficiency values for the different types of tasks in each frequency band. Fig 4A summarizes the medians observed in Fig 4B–4F. As in Fig 3, the vertical gray lines separate each type of task by their nature: panel A groups the median values for the tasks in each band while the mean value in each case is shown with a diamond.

In Fig 4A, it is shown that the highest values for global efficiency are observed in the delta band, while the lowest values are presented in the beta band. The difference between the central tendency values are at least two-fold from the beta to the delta band.

On the other hand, while the highest values for global efficiencies are obtained in the delta band, at the same time it is this band where SWN had its lowest value, closer to 1 (see Fig 3A). In contrast, the lowest SWN network is found in the delta band while the highest SWN network is found in the beta band (see Fig 3A).

In a study by Lo et al. [41], it is seen that the global efficiency increases with connection density while SWN decreases, so networks with small SWN, i.e., with values close to 1, correspond to high connection densities and, therefore, to high global efficiencies. Thus, the network corresponding to Fig 3B, can be considered a fully connected network or one in which all nodes are connected through long- and short-range connections. On other frequency bands (see Fig 4A), the values of the efficiencies are significantly lower, for both mathematical and programming tasks.

When comparing the global efficiencies for the types of tasks (mathematical and programming) in the various bands, the median values in panel A from Fig 4 show differences between

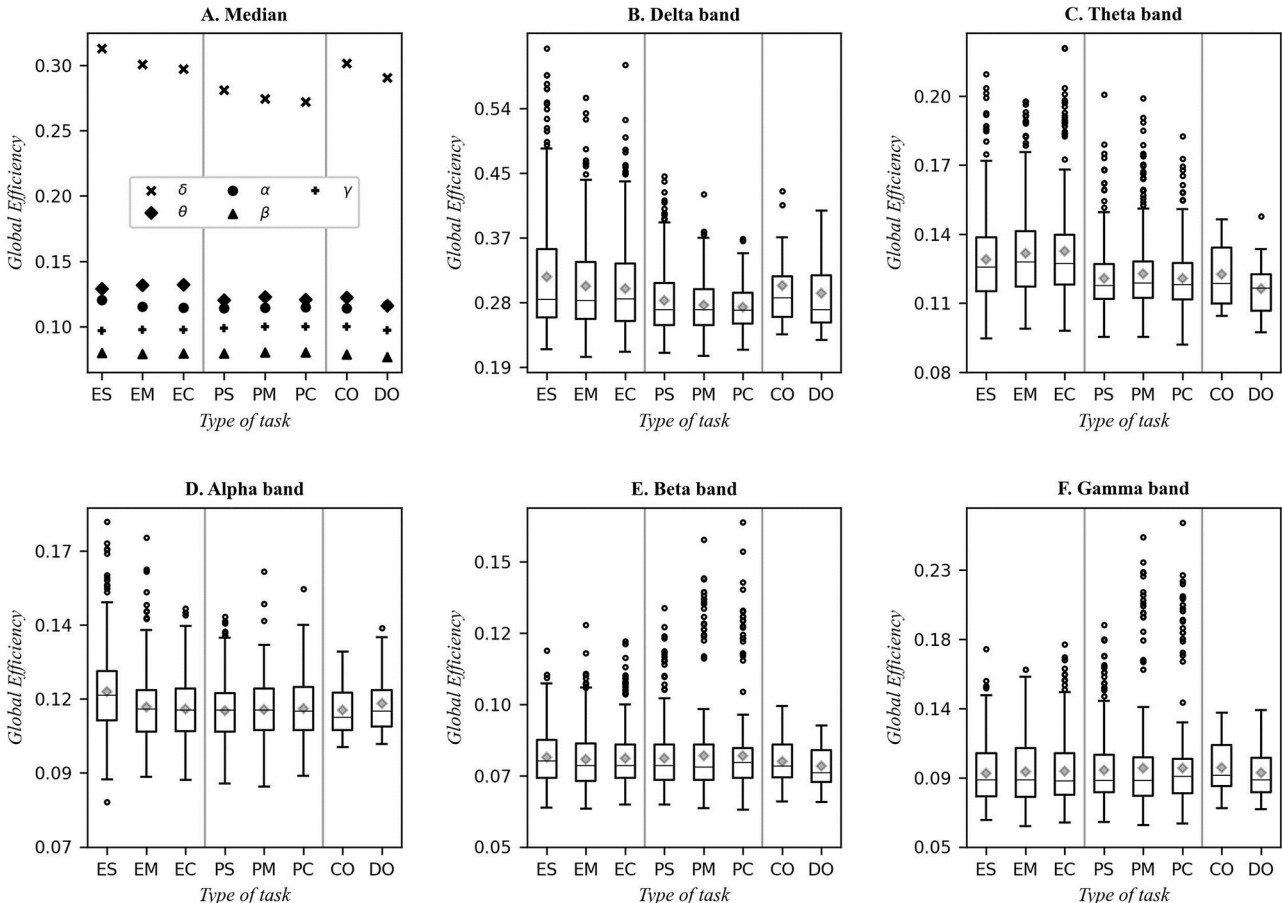

**Fig 4. Global efficiency for the delta, theta, alpha, beta and gamma frequency bands.** In A, the medians are compared, in B-F boxplots are shown for each band. The gray lines separate the tasks in to three parts: mathematical tasks (ES, EM, and EC), programming tasks (PS, PM, and PC), control condition (Cr) and subitizing (Do). Besides the median (horizontal line), the mean is shown with a diamond in each case.

those types of tasks in the delta and theta bands that are statistically significant with p-values = 0 ($10^{-6}$) for U Mann-Whitney tests. In the case of the theta band, statistical tests show that the programming tasks differ from the mathematical tasks (p-values lower than 0.05). In addition, in this band, significant differences are also observed for the simpler mathematical tasks compared to all other tasks, according to the p-values on the supporting information (see S2 Table) for the theta band and global efficiency.

Hypothesis testing on beta and gamma bands do not provide sufficient evidence to reject the null hypothesis, thus differences cannot be stated between the means.

## Local efficiency

In Fig 5B–5F, heat maps are shown in gray scale representing the values for local efficiencies on each channel and frequency bands: delta (B), theta (C), alpha (D), beta (E) and gamma (F), while in panel A from Fig 5 mean values for each band are shown. The shades of the maps range from white, representing a null local efficiency, to black, with the value of local efficiency in the (0, 0.4) range.

Vertical shadings correspond to differences in task type (mathematical or programming), while horizontal shading refers to levels of local activation.

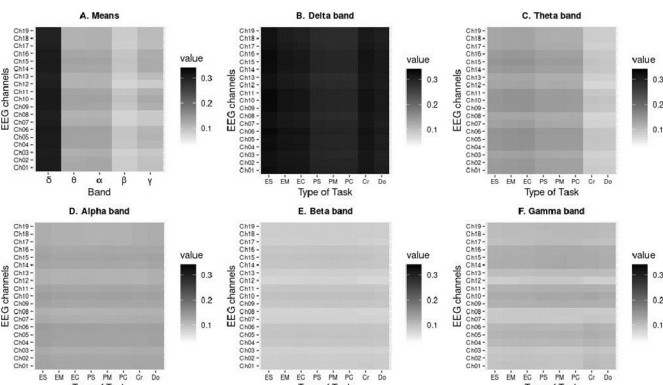

**Fig 5. Local efficiencies of each channel.** Heat maps are in gray scale, representing the local efficiencies of each channel during the development of different types of tasks for the delta (B), theta (C), alpha (D), beta (E) and gamma (F) frequency bands. The shades range from white, representing a null local efficiency, to black, which represents the maximum value of local efficiency. (A) shows the heat map for the average of local efficiencies in panels B-F.

Although the type of mathematical tasks considered in previous studies [15, 42–44] differs from the solution of first-order algebraic equations used here, common characteristics hold: the requirement of attention and memory; performance of mental tasks; number comparison, and basic inter-digit operation processes. Therefore, in this study, although the nature of the mathematical task differs from previously published works, the relationship between the delta and theta bands holds, due to the cognitive processes related to mathematical operations (see for instance [42–53]). Thereby, we can conjecture some reasons which may explain the highest local efficiencies observed in the delta (primarily) and theta bands (see Fig 5A–5C) during the solutions of equations.

The programming tasks, although arguably based on logic like the mathematical tasks, seem to have lower local efficiencies than those corresponding to the mathematical tasks on the delta and theta bands (as seen by the difference in vertical shadings between the types of tasks). This may be because, during programming tests, and due to students' level of experience, they only had to read the algorithm without performing operations to arrive at the solution. Thus, the local efficiency or effectiveness with which the information from each brain region, represented in a channel, is integrated with the information from immediate regions represented in other nearby channels is lower due to a lower execution of cognitive tasks. On the other hand higher local efficiencies in mathematical tasks in these bands may be because they not only involved the reading of equations but also involved the performance of mental arithmetic operations, thus involving greater effort.

Local efficiencies, as its name implies, are local to particular channels. The graphs on Fig 5 for panels C-F also depict differences in levels for various channels (horizontal stripes). It is of interest to determine whether there are similarities on $E_l$ levels that would determine a joint activation of these channels, which in turn, are associated with specific Brodmann regions [54, 55]. Of interest also are the similarities and differences detected during the cognitive processing in the different types of tasks.

Hypothesis testing for a particular band and type of task, fails to reject the null hypothesis or mean similarity between channels. In order to establish similarity patterns between channels in the local efficiencies, clustering techniques were used (K-means algorithm) as a method for clustering the 19 channels [56] from local efficiency data over subjects on the 20 stimuli for each of the 6 main tasks (corresponding to the 3 levels of difficulty for

mathematical and programming problems). Therefore for each band and each task we obtained 19 channel signals of length 20, and calculated the groupings of similar activation values in each instance.

Given the fact that the delta and theta bands showed the greatest differences in values amongst the two types of tasks and show differences between tasks in the global measures, the analysis is focused on those specific bands.

In determining the number of clusters to consider, a dendrogram was made in each case and the silhouette index was computed. From those experiments one notices that K = 2 or K = 3 are both suitable choices for the number of clusters (see S1 and S2 Figs), however in this work we focused on K = 3 to provide greater specificity to the groupings.

Figs 6–8 depict the channel clusters for each of the 6 tasks on the delta band that share similar cluster members. The channels that are common for the majority of clusters for each task type (all difficulty levels) but not for the other type of task have been colored black as distinguishing elements for that particular task type. We call these electrodes *differentiating elements* as they represent an exclusive activation for a particular task type. The channels in dark gray represent those that are specific to each cluster but are not differentiating elements. The channels colored in light gray are not part of the cluster. The channels grouped by a dotted gray line, correspond to the electrodes in common on all clusters.

A summary of the most prominent findings follows for the various clusters in the delta and theta bands.

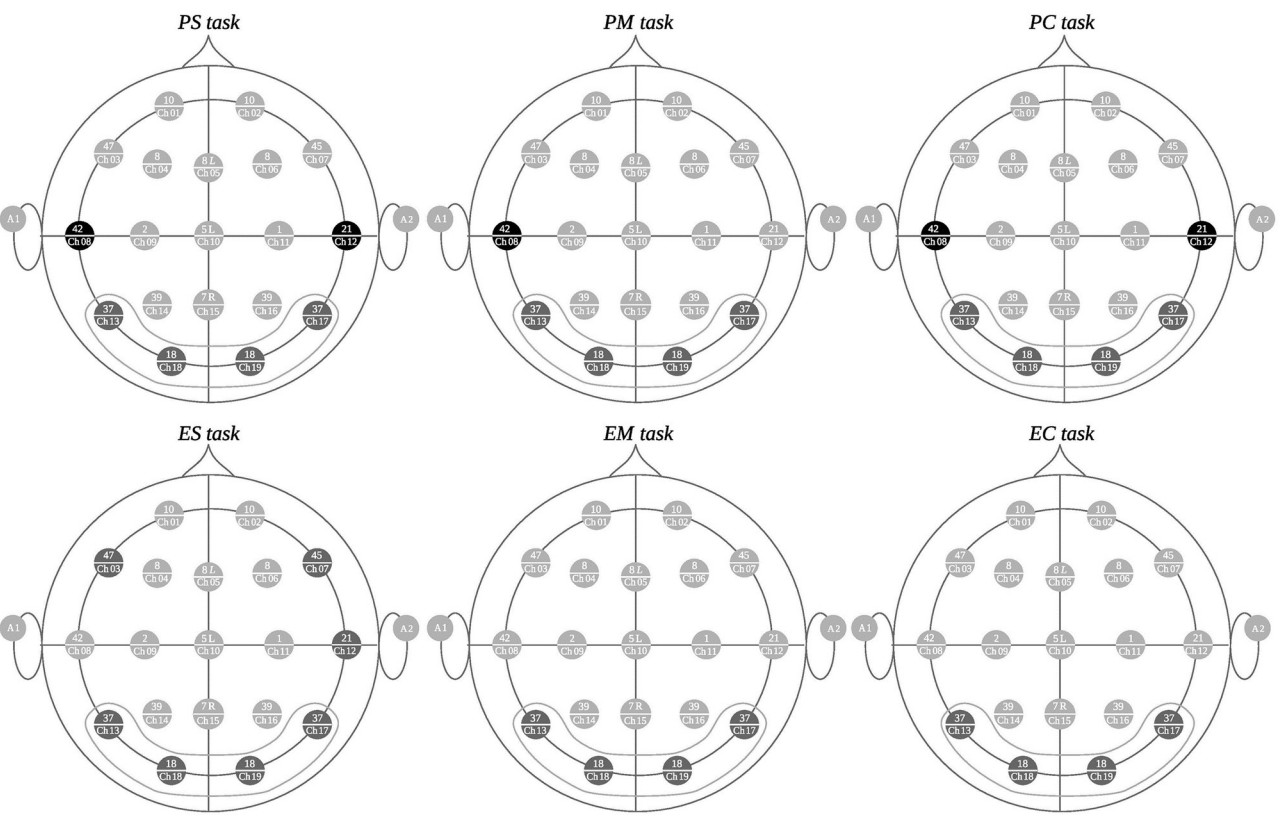

**Fig 6. First group of clusters with similar characteristics.** In the upper part, the clusters formed during programming tasks (PS, PM and PC) are observed while the corresponding mathematical tasks (ES, EM y EC) are on the lower portion.

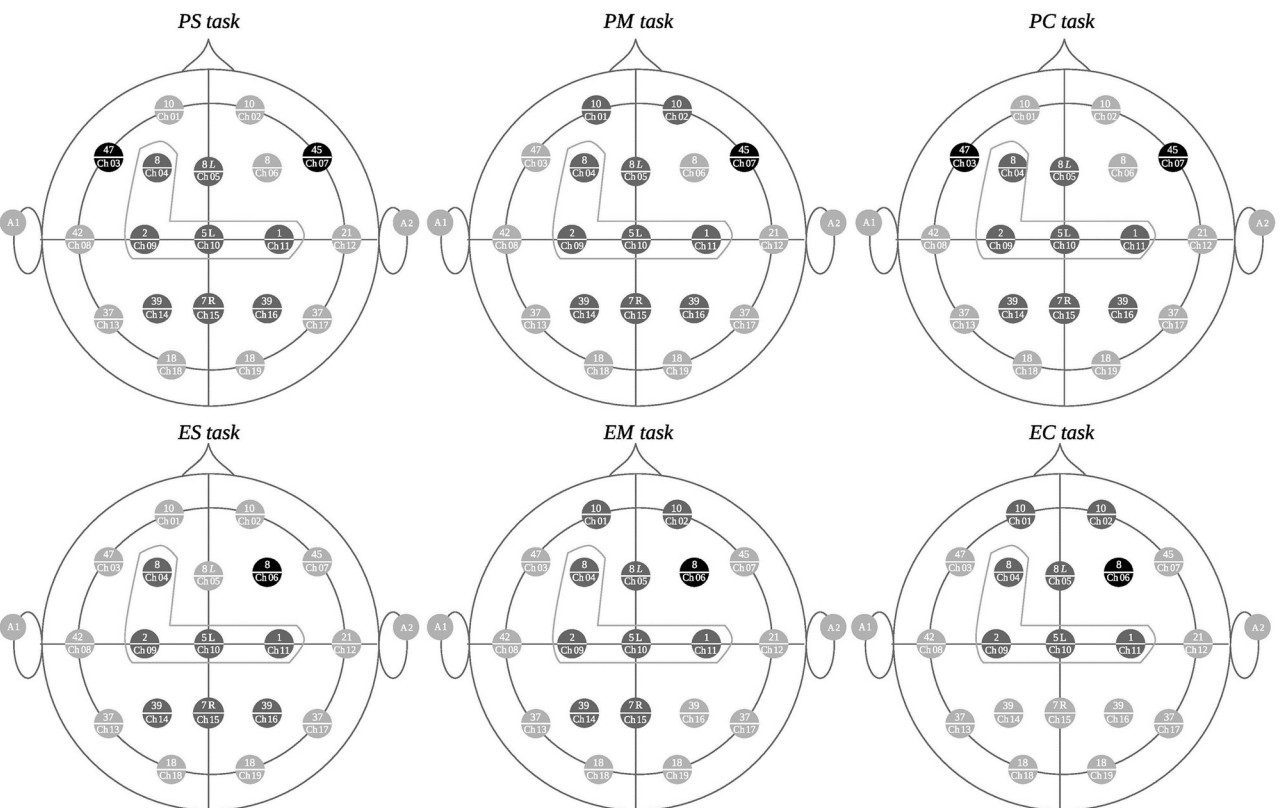

**Fig 7. Second group of clusters with similar characteristics.** In the upper part, the clusters formed during programming tasks (PS, PM, and PC) are observed while the corresponding mathematical tasks (ES, EM, and EC) are on the lower portion.

## Delta band: First cluster group

Fig 6 shows the nodes (channels), belonging to the first group of similar clusters for each task. In the upper part are the clustered channels during the execution of programming tasks (PS, PM, and PC) and in the lower part (ES, EM, and EC) for mathematical tasks. Here, channels 8 and 12 can be observed as common elements of the programming tasks to the Brodmann regions (channel 8 in *PS*, *PM*, and *PC* tasks and channel 12 in *PS* and *PC* tasks). However, they are not activated in a similar fashion for the mathematical tasks, thus establishing a difference between the activation areas observed for one activity versus the other. Channels 8 and 12 register the activity in Brodmann regions 42 and 21 respectively. Both regions are linked to the auditory process [57]. Brodmann area 42, along with area 41, form the primary auditory cortex or Heschl area, associated to various auditory processes, including the auditory working memory. Brodmann area 21 is a multimodal association area (visual, auditory and tactile) and is involved in several processes such as the prosodic integration.

Also, it is important to note that both types of tasks have in common the participation of the regions represented in channels 13, 17, 18 and 19. These are the Brodmann regions 18 and 37 in both brain hemispheres and are in the primary visual cortex, in the occipital region. Area 18 is associated with tracking visual motion patterns, visuo-spatial information processing and orientation-selective attention, among others. Area 37 is associated with semantic categorization, visual motion processing, and visual fixation, among others [58, 59]. Both of those actions take place in the mathematical and the programming tasks.

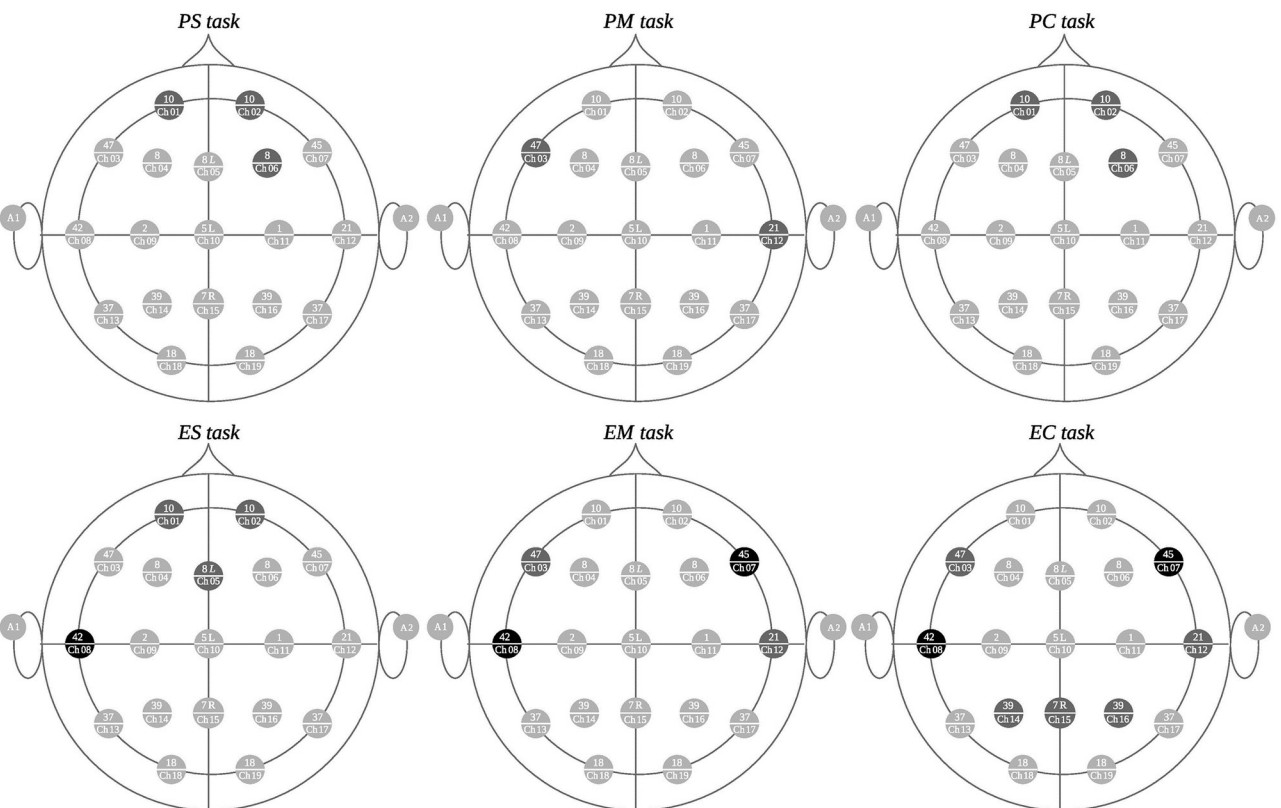

**Fig 8. Third group of clusters with similar characteristics.** In the upper part, the clusters formed during programming tasks (PS, PM, and PC) are observed while the corresponding mathematical tasks (ES, EM, and EC) are on the lower portion.

## Delta band: Second cluster group

Fig 7 shows the channels, belonging to the second group of similar clusters for each task. In this figure, the Brodmann regions represented by channels 7 (in PS task, PM task, and PC task) and 3 (in PS task and PC task) can be observed as a differentiating element of the programming tasks. Channel 7 registers the activity in region 45 (of the right hemisphere) and channel 3 registers the activity in region 47 of the left hemisphere. Brodmann area 45 is associated to grammatical processing, reasoning process and semantic memory retrieval, among others. Area 47 serves functions associated with semantic processing and encoding, word reading, working memory and deductive reasoning.

Brodmann area 8 (right hemisphere) represented by channel 6, is a distinguishing or differentiating element of the mathematical tasks (see Fig 7). This area is linked to planning, working memory, memory retrieval, visuospatial and visuomotor attention. Calculation and processing related to the uncertainty, among others [60].

In addition, it is important to point out that both types of tasks have in common the participation of regions represented by channels 9, 10 and 11. These are, Brodmann regions 1 (left hemisphere), 2 (right hemisphere) and 5 (medium line). Regions 1 and 2 carry out functions related to motor learning, touch location, fingers proprioception and the voluntary movement of the hand [61]. Region 5 is associated with the perception of personal space, motor execution, working memory (motor, visual, auditory, emotional, verbal), visuospatial memory, visuomotor attention and language processing, among others [54].

### Delta band: Third cluster group

In the third cluster group, no common patterns stand out, though regions 42 (channel 8) and 45 (channel 7), appear to be differentiating elements of the mathematical tasks as seen in Fig 8. However, there are no elements that could be considered common. Therefore, we consider that they do not show enough relevant information to infer results that indicate differences between the various types of tasks.

### Theta band: Clusters

As in the delta band, an analysis of clusters in the theta band was made. However, none of the groupings resulted in regions or channels that marked differences between mathematical and programming tasks. In fact, in each one of the three groups of clusters, a uniformity in the efficiencies of each channel group involved was observed. Therefore, the following clusters were obtained for both, the mathematical tasks and the programming tasks for the three complexity levels:

- Channels in the first group of clusters: Ch01, Ch02, Ch14 and Ch16

- Channels in the second group of clusters: Ch04, Ch05, Ch06, Ch09, Ch10, Ch11 and Ch15.

- Channels in the third group of clusters: Ch03, Ch07, Ch08, Ch12, Ch13, Ch17, Ch18 and Ch19.

One can conclude that in this band there are no observable differences in the level of activation of the channels for the two types of tasks.

## Conclusion

This work set out to find similarities and differences between the characteristics of cortical network models based on EEG during the solution of first-order algebraic equations (mathematical tasks) and the determination of the veracity of algorithms in pseudocode (programming tasks). The cortical network was characterized by three different parameters, namely the degree to which a network was a Small World Network (SWN), as well as global, and local efficiencies.

It is important to highlight that although all the results obtained in this article have been analyzed based on comparisons with previous studies, none of those studies have focused on the types of mathematical and programming tasks that we present in this article.

It was found, in accordance with previous studies, higher SWN values for the beta band (see Fig 3A and 3B), which suggests a close relationship between this coefficient and the working memory during the solution of first-order algebraic equations, and an increase in the connectivity of the network.

In terms of the similarities observed during the execution of the two types of tasks, in both cases the constructed networks with lowest Small-Worldness and the most efficient ones (mostly interconnected) were observed in the delta band. In contrast, the networks with highest Small-Worldness and the least efficient ones are observed in the beta band (see Figs 3A, 3B, 4A and 4B). These similarities reinforce the notion that the type of network and activation required for each of those tasks is similar and one may conjecture that this similarity is due to the logical thinking nature shared by both activities. In other works, this type of structure has been observed, but to the best of our knowledge, has not been contrasted with other activities of similar nature.

Only in the delta and theta bands are significant differences observed between mathematical and programming tasks, both for SWN and for $E_g$. In addition, in the theta and

alpha bands, significant differences are observed in simple mathematical tasks and with the other levels of complexity, both in the mathematical tasks and in the programming tasks.

The local efficiency is not an average or overall measure, therefore it is capable of providing greater specificity, thus the ability to exhibit the possible differences. Grouping of channels exhibiting similar structures and levels was done (in order to lower the number of variables for the comparison), and through the analysis of the various groupings, specific channels were singled out as having high activation levels or similar patterns within a same task but not shared with the other type of task. For this metric in particular it was observed that the highest local efficiency values for both, the programming tasks and the mathematical tasks, are found to a greater extent in the delta band and to a lesser extend in the theta band (See Fig 5A–5C), which are bands that have been linked with various cognitive processes associated with the performance of mathematical operations and now with programming exercises (pseudocode validation). Also, both types of tasks activated similar regions as seen in these groupings: areas 18 and 37 (channels 13, 17, 18 and 19) are common to the clusters in both types of tasks, and regions 1, 2, and 5 also share joint activation patterns. This opens the question as to whether all cognitive processes activate these channels and therefore Brodmann regions equally. However, in examining the differences between them, though the groupings formed by similar channel activation patterns during these two tasks, the subtle differences are exposed: i) Brodmann areas 21 and 42 activate during programming tasks but not during mathematical ones, and ii) Brodmann area 8 activates during the resolution of the equations in the mathematical tasks. The uniqueness of these activations is exposed by the local nature of this parameter.

It is interesting to note that Brodmann areas activated exclusively during the programming tasks are associated to auditory processes. Since the authors are unaware of previous studies in which neurophysiological responses have been explored during the development of computer programming tasks, this study is an important precedent for more rigorous studies that will allow to be established, with greater precision, the differences in the functioning of the cortical networks involved in the development of each type of task as well as a greater understanding of the underlying processes that give way to such activations.

From the results obtained in this work and the types of tasks that were applied, they show that, based on the observation and the comparison of graphs that model the function of the cerebral cortex, it cannot be stated that algorithmic thinking tasks and mathematical algebraic tasks are the same in terms of the cortical networks. It can only be stated that the cerebral cortex during the cognitive processing of both types of tasks show both similarities and differences to be explored further.

## Supporting information

**S1 Table. Statistical results for delta brainwaves.** p-values that resulted from the evaluation (asymptotic 2-tailed Mann & Witney U test) of the pairwise task differences on the parameters SWN and $E_g$.
(PDF)

**S2 Table. Statistical results for theta brainwaves.** p-values that resulted from the evaluation (asymptotic 2-tailed Mann & Witney U test) of the pairwise task differences on the parameters SWN and $E_g$.
(PDF)

**S3 Table. Statistical results for alpha brainwaves.** p-values that resulted from the evaluation (asymptotic 2-tailed Mann & Witney U test) of the pairwise task differences on the parameters SWN and $E_g$.
(PDF)

**S4 Table. Statistical results for beta brainwaves.** p-values that resulted from the evaluation (asymptotic 2-tailed Mann & Witney U test) of the pairwise task differences on the parameters SWN and $E_g$.
(PDF)

**S5 Table. Statistical results for gamma brainwaves.** p-values that resulted from the evaluation (asymptotic 2-tailed Mann & Witney U test) of the pairwise task differences on the parameters SWN and $E_g$.
(PDF)

**S6 Table. p-values: Results from Friedman test to attest the statistical differences between the 3 difficulty levels.**
(PDF)

**S1 Fig. Silhouette plots—Delta band.** Silhouette plots to select the cluster number on KMeans clustering for local efficiencies on delta band.
(TIF)

**S2 Fig. Silhouette plots—Theta band.** Silhouette plots to select the cluster number on KMeans clustering for local efficiencies on theta band .png.
(TIF)

**S3 Fig. Cluster—Delta band.** Direct results of applying K-means clustering on the delta band. Fig 6 was obtained from these results.
(TIF)

**S4 Fig. Cluster—Theta band.** Direct results of applying K-means clustering on the theta band. Fig 7 was obtained from these results.
(TIF)

## Acknowledgments

I am very grateful to the coordination of the Chemistry and Biology Department at Universidad del Norte for their permanent support in this research.

## Author Contributions

**Conceptualization:** Oscar Hernández, Minaya Villasana.

**Data curation:** Eduardo Zurek, Minaya Villasana.

**Formal analysis:** Oscar Hernández, Eduardo Zurek, Minaya Villasana.

**Funding acquisition:** Oscar Hernández, Minaya Villasana.

**Investigation:** Oscar Hernández, Eduardo Zurek, Minaya Villasana.

**Methodology:** Oscar Hernández, John Barbosa, Minaya Villasana.

**Resources:** Oscar Hernández, John Barbosa.

**Software:** Eduardo Zurek, John Barbosa.

**Supervision:** Eduardo Zurek, Minaya Villasana.

**Validation:** Minaya Villasana.

**Writing – original draft:** Oscar Hernández, Minaya Villasana.

**Writing – review & editing:** Oscar Hernández, Minaya Villasana.

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
