## [Decision Letter · Decision Letter 0]

27 Feb 2023

PONE-D-22-24455A comparative study of the cortical function during the interpretation of algorithms in pseudocode and the solution of first-order algebraic equationsPLOS ONE

Dear Dr. Hernández,

Thank you for submitting your manuscript to PLOS ONE. After careful consideration, we feel that it has merit but does not fully meet PLOS ONE’s publication criteria as it currently stands. Therefore, we invite you to submit a revised version of the manuscript that addresses the points raised during the review process.

We look forward to receiving your revised manuscript.

Kind regards,

Moo K. Chung

Academic Editor

PLOS ONE

Journal Requirements:

2. Please provide additional details regarding ethical approval in the body of your manuscript. In the Methods section, please ensure that you have specified the name of the IRB/ethics committee that approved your study."

3. Please ensure that you have specified (1) whether consent was informed and (2) what type you obtained (for instance, written or verbal, and if verbal, how it was documented and witnessed). If your study included minors, state whether you obtained consent from parents or guardians. If the need for consent was waived by the ethics committee, please include this information.

Reviewers' comments:

Reviewer's Responses to Questions

**Comments to the Author**

1. Is the manuscript technically sound, and do the data support the conclusions?

Reviewer #1: Partly

Reviewer #2: Yes

2. Has the statistical analysis been performed appropriately and rigorously? 

Reviewer #1: Yes

Reviewer #2: Yes

3. Have the authors made all data underlying the findings in their manuscript fully available?

Reviewer #1: No

Reviewer #2: Yes

4. Is the manuscript presented in an intelligible fashion and written in standard English?

Reviewer #1: Yes

Reviewer #2: Yes

5. Review Comments to the Author

Reviewer #1: This article investigates brain activity recorded from EEG using a graph theoretical approach during mathematical and programming tasks. The manuscript is very interesting and well-written. However, I am not confident about the inferences made to brain areas. EEG activity is recorded from the scalp. It is therefore unclear to me how inferences to Brodmann areas can be made. This part of the manuscript therefore needs extensive methodological description or should be removed.

The last paragraph of the introduction should conclude by clearly stating the aims and hypotheses of the manuscript, based on previous research. It is not necessary to summarize the organization of a research article as this is always predetermined.

Why was the age range of 18-21 years set as a selection criterion?

Was the EEG data pre-processed at all? I.e., ocular correction, artefact rejection etc. Otherwise some of the finding might be due to noise in the data

Please present some basic descriptives of the sample, ie. Age (mean, SD), gender etc

EEG is recorded from the scalp. How were inferences to brain areas (Brodmann areas) made? Was source localization performed? Even if this was the case, source localization is unlikely to determine precise brain areas. Please elaborate on how this analysis was conducted and valid these results are.

Reviewer #2: Motivation, novelty and conclusion

This submission has a clear and strong motivation to understand the cortical function difference, using EEG, during mathematical and computational challenges. The novelty of the paper is clear and I am happy that the conclusion can be made via a clearly designed empirical study and a relatively solid statistical analysis.

Metrics, and research methodology

The demonstration of design, shown in Figure 1, is very clear. A minor note, I am very curious about how the order of challenging matters for the final conclusion (e.g., will some level of task randomization influence the final conclusion).

I am happy to see researchers apply solid graphical analysis via metrics to define and interpret their conclusions and findings. A few minor notes:

I understand the SWN is a very useful metric to understand connectivity. However, it might be very easy to understand for most readers. Maybe we can demonstrate a few examples (either real data examples, or toy examples) to show a better visualization.

SWN is known to be very sensitive to network size. This might need further clarification. A similar metric can be L_r/L - C / C_l where r is for randomized graph and l is for lattice graph.

The definition of C_rand is not very clear – is it generated through Erdos-Renyi with a fixed number of nodes/edges or with respect to some random weighting?

Eg seems to have the same definition of C?

I am not suggesting removing the discussion of Eg and El, but it seems to be the SWN analysis is more interesting.

Statistical analysis and result

This submission uses solid statistical analysis methods to support the majority of their findings, which are mostly innovative and convincing. Some major or minor comments below.

[Major] It is not clarified in the paper how replication was performed for each sample. As there are only 16 subjects enrolled in this study, it would be helpful to understand whether the sample replication was performed and so how the sample-level heterogeneity/fixed effect influenced the research conclusion.

[Major] Authors applied pairwise non-parametric tests as evidence to show their conclusion. I am curious to understand how the multiple group anova or mixed effect generalized linear model tells us. For example, the difficulty can be a categorical variable, band can be another covariate, computing/mathematical task can be another dummy variable. A few interactions can also be useful (e.g., interaction between problem types and difficulties, etc). This can help us to gather more information related to interactions, rather than the pairwise comparison.

[Major] In the supplementary table S1 - S5, which supports the main conclusion of the paper, it includes a large amount of p-values. There is a concern of multiplicity (e.g., false positive with multiple tests). Those p-values less than 0.05 (highlighted) are very tiny (e.g., <10^-5) so I am supportive of most of the key findings but it is worth checking or at least clarifying how the result will change if we apply multiplicity adjustment.

[Minor] F3 and F4 are informative but there might be a few ways to improve. E.g., 1 is a relatively important cutoff for small world parameters, so it would be helpful to align with the same y axes range, and to clarify/interpret the result.

[Minor] I am not seeing an informative interpretation on Fig 5. However, as you find some significant differences in SWN, I am very keen to see a few demonstrations or toy examples showing how visually the connectivity differs.

6. PLOS authors have the option to publish the peer review history of their article (what does this mean?). If published, this will include your full peer review and any attached files.

Reviewer #1: No

Reviewer #2: No

---

## [Author Response · Author response to Decision Letter 0]

3 May 2023

We would like to thank the anonymous reviewers for their comments and insights, some of which found their way into the manuscript improving the overall exposition. Here we would like to address some of the issues and in some cases how these were included in the main text (and highlighted in this letter). 

Reviewer No. 1.

 1) Comment: The last paragraph of the introduction should conclude by clearly stating the aims and hypotheses of the manuscript, based on previous research. It is not necessary to summarize the organization of a research article as this is always predetermined. 

Answer: The last paragraph of the introduction was reformed to include a clear expression of the objectives instead of outlining the paper organization. Specifically, the following paragraph was added: 

“This work stems from the statement that algorithmic thinking is intimately related to mathematical thinking, thus it can be speculated that these tasks may give rise to cortical networks that share similarities in terms of the measured parameters of the cortical networks such as global and local efficiencies and smallworldness. To assess this speculation, we propose the measurement of these parameters while mathematical and programming tasks were performed to determine its characteristics, similarities, and differences between them while performing mathematical tasks and algorithmic tasks with varying levels of difficulty.” 

 2) Comment: Why was the age range of 18-21 years set as a selection criterion?

Answer: The age range between 18 and 21 years is because the recruited subjects are students of fifth semester of System’s Engineering from the university where this study was conducted. The ages of these subjects lie within this range and the selection of the semester is since the subjects must have the necessary programming abilities to complete the algorithmic tasks. A brief explanation was introduced in the article to clarify this information (see section 1.1)

“All the participants met the following selection criteria: students that dominate the programming requisites for the tasks which is typically attained after the 4th semester of System’s engenieering thus inducing an age range between 18 and 21 years , with normal or corrected vision, right-handed, not having consumed (prior to recordings) coffee, tea or other nervous system stimulant, not having drunk alcohol in the 48 hours prior to test day.”

 3) Comment: Was the EEG data pre-processed at all? I.e., ocular correction, artefact rejection etc 

Answer: Ocular correction and artefact rejection was performed through the ICA from the EEGLAB software. The signal adjustments from the EEG tick marks were performed in Matlab with in-house programming. Section 1.2 now includes this information.

 4) Comment: Please present some basic descriptives of the sample, ie. Age (mean, SD), gender etc

Answer: Age distribution for the 16 subject sample used in this study is: 1 participant was 18 years old, 9 were 19 years old, while 3 participants were 20 and 21 years old respectively.The mean age is 19.5 and the gender distribution was 2 female and 14 male subjects. This information was included in section 1.1.

 5) Comment: I am not confident about the inferences made to brain areas. EEG activity is recorded from the scalp. It is therefore unclear to me how inferences to Brodmann areas can be made .

Answer: The use of EEG electrodes placed at corresponding positions on the scalp to make inferences about the function of Brodmann's areas is a valid approach supported by numerous studies (See the references listed below). While the use of source localization algorithms can provide more precise information about the location of the underlying cortical activity, these algorithms are not necessary to establish a reliable correspondence between EEG signals and cortical function.

 • Arsalidou, M., Pawliw-Levac, M., Sadeghi, M., & Pascual-Leone, J. (2018). Brain areas associated with numbers and calculations in children: Meta-analyses of fMRI studies. Developmental cognitive neuroscience, 30, 239-250.

 • Damiani, D., Nascimento, A. M., & Pereira, L. K. (2020). Cortical Brain Functions–The Brodmann Legacy in the 21st Century. Arquivos Brasileiros de Neurocirurgia: Brazilian Neurosurgery, 39(04), 261-270.

 • Koechlin, E., & Hyafil, A. (2007). Anterior prefrontal function and the limits of human decision-making. Science, 318(5850), 594-598.

 • Michel, C. M., Murray, M. M., Lantz, G., Gonzalez, S., Spinelli, L., & Grave de Peralta, R. (2004). EEG source imaging. Clinical neurophysiology, 115(10), 2195-2222. doi: 10.1016/j.clinph.2004.06.001

 • Vrba, J., & Robinson, S. E. (2001). Signal processing in magnetoencephalography. Methods, 25(2), 249-271. doi: 10.1006/meth.2001.1238

No source localization was done but common practices were followed to make it easier to understand.

Reviewer No. 2.

 6) Comment: I am very curious about how the order of challenging matters for the final conclusion (e.g., will some level of task randomization influence the final conclusion)

Answer: Similar studies begin the testing with the control task of white cross observation over black followed by subitizing task. Thereafter, the order of the tasks used here is similar to those used in neuroscience and experimental psychology, from simplest to hardest. These decisions exclude any possible variation of the results seen here due to a change in task ordering. For the same task the order of presentation of the items was random, but the decision to start with the mathematical testing followed by the programming tasks was arbitrary and we consider that no significant change would have been observed if the order were reversed. This is a hypothesis that could be tested in further follow-up studies but are not addressed here. 

 7) Comment: I understand the SWN is a very useful metric to understand connectivity. However, it might be very easy to understand for most readers. Maybe we can demonstrate a few examples (either real data examples, or toy examples) to show a better visualization.

Answer: One way to better understand the SWN metric is to examine the transition of a regular network into a random network. The figure below shows a computational model to obtain small world networks while increasing the probability 𝑝 of randomly connecting nodes. This is, if we connect some nodes randomly from the regular network, C is increased while L remains approximately the same as a small world network. 

We hope this helps to clarify the matter; however it has not been included into the manuscript to prevent extending the article further, however references to these metrics have been included in the text (section 1.4) when defining the measure. 

The following references may help readers with a better understanding of this metric and its importance for network characteristics: 

 • Bassett DS, Bullmore E. Small-world brain networks. The Neuroscientist: A Review Journal Bringing Neurobiology, Neurology and Psychiatry. 2006;12(6):512–523. doi:10.1177/1073858406293182.

 • Humphries, Mark D., and Kevin Gurney. "Network ‘small-world-ness’: a quantitative method for determining canonical network equivalence." PloS one 3.4 (2008).

 • Bassett, Danielle S., and Edward T. Bullmore. "Small-world brain networks revisited." The Neuroscientist 23.5 (2017): 499-516.

 8) Comment: SWN is known to be very sensitive to network size. This might need further clarification. A similar metric can be L_r/L - C / C_l where r is for randomized graph and l is for lattice graph. 

Answer: In regard to the SWN sensibility with respect to the size of the network, in Humphries M.D. and Gurney K. (2008) it was shown for social, technological and biological networks that SWN increases linearly with the size of the network. This reference was included in the article and referred to in section 1.4 to emphasize this dependency.

Network ‘Small-World-Ness’: A Quantitative Method for Determining Canonical Network Equivalence.

The reviewer suggests an alternative for SWN. Such an alternative is plausible, however given the fact that many previous works use SWN as a network measure we think that this can help readers to relate to other published works.

 9) Comment: The definition of C_rand is not very clear – is it generated through Erdos-Renyi with a fixed number of nodes/edges or with respect to some random weighting?

Answer: The random graphs were created using the HERMES software which states: “The graph created is entirely random, with the probability of connection tested for each pair of vertices (Erdos-Renyi random graph). Only single edges and no self-loops are allowed - hence, P is adjusted to account for the slight reduction in the number of maximum edges”. This explanation is included in section 1.4

 10) Comment: It is not clarified in the paper how replication was performed for each sample. As there are only 16 subjects enrolled in this study, it would be helpful to understand whether the sample replication was performed and so how the sample-level heterogeneity/fixed effect influenced the research conclusion.

Answer: Replication was not possible due to the costs involved in the experiment both in money and time. However, we observed that the standard deviation for the various metrics is reasonably low for 20 randomized items within each task so there is some homogeneity for each subject.

 11) Comment: Authors applied pairwise non-parametric tests as evidence to show their conclusion. I am curious to understand how the multiple group anova or mixed effect generalized linear model tells us. For example, the difficulty can be a categorical variable, band can be another covariate, computing/mathematical task can be another dummy variable. A few interactions can also be useful (e.g., interaction between problem types and difficulties, etc). This can help us to gather more information related to interactions, rather than the pairwise comparison.

Answer: Statistical test show that the data does not follow a normal distribution, therefore anova or manova should not be applied as normality is an underlying hypothesis for the anova. However, in its place and thanks to this comment. The Friedman test was used to determine if one could differentiate the 3 levels of difficulty for each type of task. This is now found in the supplementary materials. We did not evaluate differentiation of difficulty level mixing types of tasks, since we cannot presume that programming and mathematical tasks are of similar nature and is the objective of this manuscript. 

Furthermore, we considered that given the differences in ranges for SWN, Eg and El, and the existing literature where these metrics are reported segregated by bands, it doesn’t make sense to consider all the bands simultaneously with regards to the two types of tasks and the difficulty levels. 

The Friedman tables are added in the supplementary materials and are referenced in section 1.5.

The supplementary material also includes results from Friedman test to attest the statistical differences between the 3 difficulty levels complementary to the U-Mann Whitney test. 

 12) Comment: In the supplementary table S1 - S5, which supports the main conclusion of the paper, it includes a large amount of p-values. There is a concern of multiplicity (e.g., false positive with multiple tests). Those p-values less than 0.05 (highlighted) are very tiny (e.g., <10^-5) so I am supportive of most of the key findings but it is worth checking or at least clarifying how the result will change if we apply multiplicity adjustment.

Answer: The experiment was carried out with 6 different tasks and 20 exercises in each task (ES, EM, EC, PS, PM and PC), in addition to two control activities (CO and DO). Each of the exercises was performed by 16 volunteer participants, generating a total of 320 samples per task. The power of the Mann-Whitney U test is calculated as indicated in [Sample Size Calculator. Available from: https://homepage.univie.ac.at/robin.ristl/samplesize.php?test=wilcoxon], with parameters of a sample size per group of 320, the obtained p-values and a significance level of 0.05 (alpha two-sided 0.05), and as a result a power greater than 0.99 is obtained.

The format of the results shown in tables S1, S2, S3, S4 and S5 in supplementary materials were changed to scientific notation in order to improve readability.

 13) Comment: F3 and F4 are informative but there might be a few ways to improve. E.g., 1 is a relatively important cutoff for small world parameters, so it would be helpful to align with the same y axes range, and to clarify/interpret the result.

Answer: It was noted that in Figures 3 and 4 the range for each picture panel is different. When placing them on a unifying range, in many cases these were no observable differences in the distributions or boxplots at the overall scale and range because information appeared “lumped” together, but the differences became evident when depicting the values in their natural range. However, a unifying range is shown in panel A for both figures where the medians are plotted and differences between the ranges per bands could be evident. 

 14) Comment: I am not seeing an informative interpretation on Fig 5. However, as you find some significant differences in SWN, I am very keen to see a few demonstrations or toy examples showing how visually the connectivity differs.

Answer: Figure 5 shows heatmaps for El and not for SWN which was addressed in the previous comment. The El heatmaps of figure 5 and the information contained within are further explored using clustering techniques (figures 6, 7 and 8 in section 2.3). The other comment regarding toy examples to demonstrate connectivity was addressed in point 7) in this document. 

Please do not hesitate to contact us if further clarifications are required.

---

## [Decision Letter · Decision Letter 1]

12 Jun 2023

A comparative study of the cortical function during the interpretation of algorithms in pseudocode and the solution of first-order algebraic equations

PONE-D-22-24455R1

Dear Dr. Hernández,

We’re pleased to inform you that your manuscript has been judged scientifically suitable for publication and will be formally accepted for publication once it meets all outstanding technical requirements.

Kind regards,

Fabio Rapallo, Ph.D.

Academic Editor

PLOS ONE

Additional Editor Comments (optional):

Both reviewers agree that the authors have carefully addressed the concerns on the first version.

Reviewers' comments:

Reviewer's Responses to Questions

**Comments to the Author**

1. If the authors have adequately addressed your comments raised in a previous round of review and you feel that this manuscript is now acceptable for publication, you may indicate that here to bypass the “Comments to the Author” section, enter your conflict of interest statement in the “Confidential to Editor” section, and submit your "Accept" recommendation.

Reviewer #1: All comments have been addressed

Reviewer #2: All comments have been addressed

2. Is the manuscript technically sound, and do the data support the conclusions?

Reviewer #1: Yes

Reviewer #2: Yes

3. Has the statistical analysis been performed appropriately and rigorously? 

Reviewer #1: Yes

Reviewer #2: Yes

4. Have the authors made all data underlying the findings in their manuscript fully available?

Reviewer #1: Yes

Reviewer #2: Yes

5. Is the manuscript presented in an intelligible fashion and written in standard English?

Reviewer #1: Yes

Reviewer #2: Yes

6. Review Comments to the Author

Reviewer #1: The authors have adequately addressed by concerns and feedback. I believe the manuscript is now in a format suitable for publication.

Reviewer #2: Thanks for carefully addressing my comments. Most of my comments are well addressed and some pushbacks are reasonable.

LGTM to accept.

7. PLOS authors have the option to publish the peer review history of their article (what does this mean?). If published, this will include your full peer review and any attached files.

Reviewer #1: No

Reviewer #2: No

---

## [Editor Report · Acceptance letter]

16 Jun 2023

PONE-D-22-24455R1 

A comparative study of the cortical function during the interpretation of algorithms in pseudocode and the solution of first-order algebraic equations 

Dear Dr. Hernández:

I'm pleased to inform you that your manuscript has been deemed suitable for publication in PLOS ONE. Congratulations! Your manuscript is now with our production department. 

Kind regards, 

on behalf of

Dr. Fabio Rapallo 

Academic Editor

PLOS ONE